

# ICESHEET 1.0: A program to produce paleo-ice sheet models with minimal assumptions

Evan J. Gowan[1,2,3], Paul Tregoning[3], Anthony Purcell[3], James Lea[1,2], Oscar J. Fransner[4], Riko Noormets[4], and J. A. Dowdeswell[5]

[1]Department of Physical Geography, Stockholm University, Stockholm, Sweden
[2]Bolin Center for Climate Research, Stockholm, Sweden
[3]Research School of Earth Science, The Australian National University, Canberra, Australia
[4]Department of Arctic Geology, The University Center in Svalbard (UNIS), P.O. Box 156, 9171 Longyearbyen, Norway
[5]Scott Polar Research Institute, Cambridge, UK

*Correspondence to:* Evan J. Gowan (evangowan@gmail.com)

**Abstract.** We describe a program that produces paleo-ice sheet models using an assumption of steady state, perfectly plastic ice flow behaviour. It incorporates three input parameters: ice margin, basal shear stress and basal topography. Though it is unlikely that paleo-ice sheets were ever in complete steady-state conditions, this method can produce an ice sheet without relying on complicated

and unconstrained parameters such as climate and ice dynamics. This makes it advantageous to use in glacial-isostatic adjustment ice sheet models, which are often used as input parameters in global climate modelling simulations. We test this program by applying it to the modern Greenland Ice Sheet and Last Glacial Maximum Barents Sea ice sheet and demonstrate the optimal parameters that balance computational time and accuracy.

**1    Introduction**

Modelling past ice sheets is a complex task, due to the large number of parameters that can affect their growth and retreat. For example, Tarasov et al. (2012) presented a glacial systems model that contained 39 parameters that could be tuned, which included climatology, Earth rheology, ice physics and margin chronology. Many of these parameters are poorly constrained by available ob-

servations. In particular, past climate is often parameterized based on ice core data from Greenland and Antarctica, or reconstructions from speleothems that are located far from where the ice sheets existed.

Since past climatic parameters are generally only well characterized in areas outside of where paleo-ice sheets existed, ice sheet models that are independently determined using evidence of

glacial-isostatic adjustment (GIA) are often used in paleo-climate simulations (*e.g.* Braconnot et al., 2007, 2012). One of the most commonly used GIA based models of glaciation is the ICE-xG series (*e.g.* Peltier, 2004; Peltier et al., 2015). They produce configurations of ice sheets that minimize the




misfits of geodetic and relative sea level data, with limited regard to the physical realism of the ice
sheet itself. Another commonly used model is the ANU model (*e.g.* Lambeck et al., 2010), which
was developed using an assumed peak ice elevation at the center of ice sheets, and using a parabolic
ice profile to the margins. In their formulation, each flowline ray allowed to have different basal shear
stress values, but is less flexible in regards to the direction of the flowline, and spatial variability in
basal shear stress along it.

The method presented in this paper is a numerical program that produces a physically realistic ice
sheet configuration while taking into account changes in basal shear stress and topography, while
being simple enough that it does not depend on numerous parameters with large uncertainties. The
model is based on the assumption of perfectly plastic, steady state ice conditions. It allows for the
rapid determination of paleo-ice sheet configurations, which is desirable when matching observa-
tions of GIA. We present an example application of this software to the Barents Sea Ice Sheet, a
relatively short lived portion of the Eurasian Ice Sheet complex. We also apply the model to the
contemporary Greenland ice sheet to provide an indication of how well the model is capable of re-
constructing a known ice sheet geometry. This software has also been used to produce a model of
the full deglacial cycle of the western Laurentide Ice Sheet (Gowan et al., in review).

## 2 Methodology

### 2.1 Theory

The ice sheet modelling software is based on the assumption that ice rheology adheres to perfectly
plastic, steady-state conditions (*i.e.* ignoring lateral shear stresses, and assuming that the ice surface
is not dynamically changing). The two-dimensional form of this theory was derived by Nye (1952),
and neglects variability in topography and longitudinal changes in stress. In this equation, the ice
surface gradient is directly related to the strength of the ice-bed interface, or basal shear stress. The
basal shear stress is related to a number of factors, including basal geology, sediment thickness and
strength, hydrology, temperature and bed roughness.

$$\frac{dE}{ds} = \frac{\tau_o}{\rho_i g H} \qquad (1)$$

The ice surface elevation is $E$, $s$ is the distance along ice flowline profile, $\tau_o$ is the shear stress at
the base of the ice sheet, which balances the driving stress, $\rho_i$ is the density of ice, $g$ is the gravity
at the Earth's surface, and $H$ is the ice thickness. If the distance from the ice sheet margin to the
centre of the ice sheet is known, then the thickness along the profile between the two points can be
calculated using the following formula (Cuffey and Paterson, 2010).

$$H^2 = \frac{2\tau_o}{\rho_i g}[L - x] \qquad (2)$$



In this equation, $L$ is the distance between the margin and centre of the ice sheet, and $x$ is the distance from the centre. Though this equation is simple, it can be used to make a rough estimate of the thickness of ice sheets (Cuffey and Paterson, 2010). Eq. 2 was used to create the ANU ice sheet model (*i.e.* Lambeck et al., 1998, 2006, 2010). The weakness of using this equation is that the center of the ice sheet has to be assumed a-priori. It also does not take into account changing basal shear

stress conditions or changes in topography.

In order to overcome problems with spatial changes in basal topography and shear stress, in addition to the uncertainties in the location of the ice sheet center, Reeh (1982) and Fisher et al. (1985) presented expanded version of Eq. 1 that allows for changes in the direction of the flowline. The equation becomes the following partial differential equation.

$$\left(\frac{dE}{ds}\right)^2 = \left(\frac{\partial E}{\partial x}\right)^2 + \left(\frac{\partial E}{\partial y}\right)^2 \tag{3}$$

The coordinate system is set up so that $x$ points towards the center of the ice sheet, and $y$ is parallel to the margin. Presented in the notation used by Reeh (1982), Eq. 1 is substituted into the left side of Eq. 3, with the ice thickness represented in terms of ice surface elevation and basal topography elevation $B$, and substituting in a characteristic thickness, $H_f = \tau_o/\rho_i g$.

$$\left(\frac{H_f}{E - B}\right)^2 = \left(\frac{\partial E}{\partial x}\right)^2 + \left(\frac{\partial E}{\partial y}\right)^2 \tag{4}$$

The above equation describes the change in ice thickness over an arbitrary surface. This partial differential equation can be solved by the method of characteristics. The $x$ and $y$ partial derivatives in Equation 4 are substituted by $p = \partial E/\partial x$ and $q = \partial E/\partial y$, then rearranged in terms of $p$.

$$p = \sqrt{\left(\frac{H_f}{E - B}\right)^2 - q^2} \tag{5}$$

The solution to the partial differential equation then becomes three ordinary differential equations that are solved simultaneously, using the method of characteristics (Reeh, 1982).

$$\frac{dy}{dx} = \frac{q}{p} \tag{6}$$

$$\frac{dE}{dx} = \frac{p^2 + q^2}{p} = \frac{H_f^2}{(E - B)^2 p} \tag{7}$$

$$\frac{dq}{dx} = \frac{(p^2 + q^2)(\partial B/\partial y - q)}{p(E - B)} = \frac{H_f^2}{p(E - B)^3}\left(\frac{\partial B}{\partial y} - q\right) \tag{8}$$



Fisher et al. (1985) expanded Equation 8 to allow for changes in basal shear stress (in terms of the characteristic thickness, $H_f$).

$$\frac{dq}{dx} = \frac{H_f^2}{p(E-B)^3} \left( \frac{\partial B}{\partial y} - q \right) + \left( \frac{H_f}{p(E-B)^2} \right) \frac{\partial H_f}{\partial y} \tag{9}$$

These equations can be solved by numerical integration to determine the course and gradient of an ice flowline.

## 2.2  Modelling procedure

In order to solve the Eqs. 6-8, initial values for $E$, $y$ and $q$ are required. Starting model calculation at the margin is convenient from the perspective of reducing a-priori assumptions on ice distribution, though it leads to a singularity because the ice thickness is zero ($E = B$). Consequently, the value of $E$ at the margin must be set to be a nominal value (in the sample problems presented in this study,

1 m). Although the actual thickness of ice near the margin may be as high as tens of metres, the choice of starting value will not have a large effect on the final model. For instance, the distance from the margin required in Eq. 2 to reach 10 m from a starting value of 1 m, and a low basal shear stress value (5 kPa) is 90 m, substantially smaller than the uncertainty in the margin location for paleo-ice sheets (Clark et al., 2012; Gowan, 2013; Hughes et al., 2016). For simplicity, the value of

$q$ is defined to be zero at the margin. This can be justified because near the margin the value of term $H_f/(E-B)$ will dominate Eq. 5 in the defined coordinate system.

The ice sheet model is calculated in a piece-wise manner (see Fig. 1 for an illustration of the steps involved). The ice flowline calculation is initiated at intervals along the margin, which are user defined. The flowline calculation proceeds until it reaches a particular elevation (a user defined

contour interval), at which point the program checks to see if any flowlines cross over, or if a saddle point in the ice sheet has been reached. A sequential list of the modelling steps is given below.

1. All parameters (ice sheet margin, shear stress map, topography map) are converted from geographical coordinates to a Cartesian coordinate system prior to the execution of the program.

2. Estimates of the basal shear stress for the area of interest are read into the program. The

shear stress values must be adjusted for each time interval to produce an appropriate ice sheet configuration.

3. The basal topography data for the area of interest are read in. For the first iteration of ice sheet model development, it uses modern topography or topography adjusted for changes in global mean sea level (in practice, it has limited impact on the final result to change the initial sea

level, even with predominantly marine based ice sheets). In subsequent iterations, it includes a component of glacial-isostatic adjustment to take into account the fact that the ice sheet will deform the Earth, and that the ice sheets will cause changes to sea level. The modified





topography is calculated before running the ice sheet program. In the Barents Sea Ice Sheet sample problem, we use the CALSEA program to calculate GIA (Nakada and Lambeck, 1987; Lambeck et al., 2003).

4. The program reads in the margin, and defines locations along the perimeter where the flow-line calculation initiates. The minimum distance along the margin between where flowline calculation is initiated is user-defined. The program defines the initial direction of flow to be perpendicular to the margin, away from the centre of the ice sheet.

5. The margin is set to have an initial ice thickness of 1 m. If the margin is located where the topography is below sea level, it is assumed that the margin corresponds to the grounding line of the ice sheet. A conservative estimate of the thickness of ice at this point is set to $H = -B(1 - \rho_{seawater}/\rho_{ice})$, where $\rho_{seawater}$ is the density of sea water and $\rho_{ice}$ is the density of ice, which is the thickness of ice corresponding to the equivalent mass of the water column at that point. There is a check to make sure that the ice surface slope between adjacent points on the boundary is not too steep for the given basal shear stress values. If it is, the ice thickness at the point with the lower elevation is increased. This check is only done where $B < 0$.

6. The calculation of ice elevation contours is a recursive process. If the contour crosses over itself (signifying a saddle on the surface of the ice sheet), the contour polygon is split, and the calculation is continued as separate polygons (*e.g* see point 6 in Fig. 1).

7. The program searches for points on the contour that are below the next contour elevation. It then calculates the flowline by numerical integration of Eqs. 6-8, using the Runge-Kutta method (Press, 1992). When it reaches the next contour elevation, the calculation stops.

8. If the flowline calculation cannot reach the next contour elevation, which happens when the topography is too steep ($H \rightarrow 0$), the point is flagged and not included in the next contour (*e.g* see point 1 in Fig. 1).

9. If the flowline direction changes sufficiently so that $q \geq H_f/(E - B)$ (*i.e.* $p$ approaches zero), the local coordinate system is rotated so that $p$ is in the direction of maximum flow.

10. If the calculated flowline goes outside the last calculated contour polygon, it is flagged and the point is not included in the next contour. This happens when the ice surface is near its peak height. This can also happen in areas where there is a sudden change in topography or basal shear stress, which causes a deflection in the flowline direction.

11. After the flowlines are calculated for each applicable point along the polygon, the program checks to see if any of the calculated flowlines cross over. Offending crossovers are eliminated



using a motorcycle algorithm (*e.g.* Vigneron and Yan, 2014). The eliminated flowlines are flagged and not included in the next contour.

12. At this point, an initial polygon of the next elevation contour can be constructed. This is
    checked to ensure that it is a simple polygon (*i.e.* a polygon that does not cross over itself).
If it is not, then the program breaks it into several polygons, and determines whether they
     represent domes (ice gradient is increasing towards the centre of the polygon) or saddles (the
     ice gradient is decreasing towards the centre of the polygon). Where a saddle is identified, it is
     determined to have reached its peak elevation and is eliminated from subsequent calculations.

13. The ice elevation and thickness for all points on a valid polygon (including flagged points) are
written to file.

14. The polygon is resampled using the user-defined distance interval. There is also a check using
    Eq. 2 to estimate the distance to the next contour. If the difference in estimated distance be-
    tween adjacent points is greater than the user defined distance threshold, additional points are
    included. This process excludes flagged points, and may incorporate basal topographic highs,
where flowline calculation will not be initiated.

This process is repeated for each time interval of interest. After calculation of the ice model, the calculated elevation values are averaged into a grid to be used as input for a GIA calculation program. The grid is created using a continuous smoothing algorithm, which is part of Generic Mapping Tools (Smith and Wessel, 1990).

## 3 Sample model - Greenland Ice Sheet

### 3.1 Setup

The Greenland Ice Sheet serves as a good example of the capabilities of the ICESHEET program. The basal topography under the ice sheet is an observationally constrained, mass continuity based inversion of the contemporary ice thickness (Morlighem et al., 2014). Reeh (1982) modelled the
Greenland Ice Sheet reasonably well using the methodology explained earlier using a constant basal shear stress of 90 kPa. Since ICESHEET can have spatially variable basal shear stress and account for variable topography, it is possible to refine this. Advances in remote sensing over the last 30 years also allow a more accurate comparison to contemporary topography.

The goal of this example is to determine the misfit between the ICESHEET modelled ice surface
topography and the contemporary ice sheet using a methodology analogous to the reconstruction of a paleo-ice sheet. The input grounded ice margin and basal topography data come from the IceBridge BedMachine Greenland, Version 2 dataset (Morlighem et al., 2014, 2015). The basal shear stress value domains were designed the same way as a paleo-ice sheet would be constructed. The domains



were constructed purely on the basis of basal topography (Fig. 2), since information on basal geol-
ogy is limited. They were predominantly divided into areas of rugged topography (*i.e.* mountainous
regions), flat lying areas, and fjords. There intentionally was no attempt to divide it on the basis of
modern ice flow patterns, given that it may not be possible to deduce them for a paleo-ice sheet.
The shear stress values in the domains were adjusted iteratively in order to try to match the observed
ice surface topography. In a paleo-ice sheet, it will not be possible to know what the ice surface
topography was a-priori. In that case, other sources of data (*i.e.* GIA) must be used as the basis for
the reconstruction.

### 3.2   Results

The resulting model is shown in Fig. 2. For comparison purposes, the ice sheet is averaged into a
25 km grid. The modelled ice sheet surface topography has an average difference of-37±2 m (within
200 m of the true topography for most of the ice sheet). The largest errors occur in places where there
are narrow ice streams, which could not be parameterized using the coarse resolution shear stress
domains. In general, the shear stress values are highest in the mountainous regions in southeastern
Greenland. The basal shear stress is lowest in the center of the ice sheet, likely reflecting the flat-lying
basal topography.

## 4   Sample model - Barents Sea Ice Sheet

### 4.1   Setup

The Barents Sea Ice Sheet was predominantly marine-based, and likely formed by the merging of
isolated ice caps over Svalbard, Franz Josef Land, Novaya Zemlya and the Scandinavia Ice Sheet
(Ingólfsson and Landvik, 2013). The hypothesis explaining the glaciation of the entire Barents Sea
is that GIA warped the floor of the Barents Sea upwards, favouring the formation of grounded ice. At
the Last Glacial Maximum (LGM) (about 20 ka), the ice sheet covered the entire continental shelf
region west of Novaya Zemlya (Ingólfsson and Landvik, 2013). The extent was probably limited in
the Kara Sea east of Novaya Zemlya, compared to the mid-Weichselian (45-55 ka) glaciation. At
the LGM, the ice thickness was likely greatest to the east of Svalbard, on the basis of the pattern of
paleo-sea level reconstructions (Lambeck, 1995).

In this sample problem, the ice sheet extent is taken as the "most likely" configuration at 20 ka
from the DATED project (Hughes et al., 2016). Since the Barents Sea Ice sheet merged with the
Scandinavian Ice Sheet at the LGM, the margin is cut off far enough south so that the northern part
of the Scandinavian Ice Sheet is sufficiently represented. The basal topography used in this problem
is from IBCAO (Jakobsson et al., 2012). The basal topography of Svalbard takes into account the
thickness of modern ice cover. There is no published information on the thickness of ice on No-
vaya Zemlya, so we use contemporary ice surface topography. The basal shear stress was initially



parameterized on the basis of topography and bedrock geology. The values were adjusted in order to produce an ice thickness distribution that is similar to the GIA based ANU model (Lambeck, 1995;
Lambeck et al., 2006, 2010). Exact matching of ice thickness in the sample problem to the ANU model was not attempted, since it is of low resolution, and has a different margin configuration to that of Hughes et al. (2016). Specifically, it is less extensive along the Bear Island Trough. In order to approximate the ice thickness from the ANU model, the basal shear stress was set to be high along the northern part of the ice sheet, and relatively low in the southern Barents Sea. Both the topography
and basal shear stress values are sampled at 5 km (Fig. 3).

This purpose of this test is to demonstrate that GIA has an impact on the ice sheet configuration. This test only includes the Barents Sea Ice Sheet for the calculation of GIA. In a full simulation, it is necessary to include the effects of far field ice sheets, and realistic ice sheet growth and decay.

### 4.2   Resolution test

In order to test the optimal parameters for producing ice sheet configurations, a series of tests with different distance and contour intervals were performed. This test involved using modern topography minus the approximate 133 m reduction in global mean sea level at 20 ka (Fig. 3, Lambeck et al., 2014). The shear stress and basal topography values are shown in Fig. 3. Fig. 4 shows how changes in basal shear stress and basal topography affect the modelled ice sheet. The spacing between contours
is greater in areas of low basal topography and shear stress, which replicates ice flow from areas of high to low basal topography, and around barriers that resist ice flow. The results of this test are shown in Table. 1.

The program execution time largely depends on the chosen sampling interval along the contour polygons. The reference ice sheet configuration used a distance interval of 1 km, and a contour
interval of 10 m (Fig. 5). Unsurprisingly, considering the 5 km resolution grid, all tests using distance intervals 5 km or less produced nearly identical configurations, as they captured the details of the grids. Using a contour interval of 20 m gives almost the same result as as 10 m, with diminishing accuracy when increased above this, without significant reductions in execution time. The optimal parameters for matching the reference configuration and fast execution time are a 5 km spacing and
20 m contour interval (Table. 1). Increasing the distance parameter decreases the execution time, but is unable to match the reference model, particularly in the mountainous regions of Svalbard and Scandinavia. There is a tendency towards overestimating the ice thickness when the initiation distance is larger than 5 km (Fig. 5). During the initial phases of GIA based ice model development, it may be prudent to decrease the resolution of the grids to quickly determine an estimate of basal
shear stress, then increase the resolution when refinement is necessary.

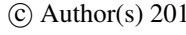



### 4.3 GIA test

When an ice sheet grows, the basal topography is modified by GIA, which will significantly impact the Barents Sea Ice Sheet example. Therefore, in order to obtain an accurate characterization of the ice sheet surface topography and thickness, it is necessary to re-run the simulation with the modified basal topography. The Earth model used in this sample problem is spherically symmetric and includes a 90 km thick elastic lithosphere, $4 \times 10^{20}$ Pa s upper mantle viscosity and $10^{22}$ Pa s lower mantle, which is in the range of best fitting models for this region (Lambeck et al., 2010). The distance interval used is 5 km and contour interval is 20 m. Since there is a viscous component of the response, the ice sheet is allowed to grow linearly from 30 ka (when glaciation in the Barents Sea is presumed to be similar to present, Mangerud et al., 1998) to 20 ka, then linearly decrease back to present levels at 10 ka. After the first iteration of GIA, the ice sheet contribution to global mean sea level is subtracted to determine the Earth deformation. When combined with the actual global mean sea level at this time (-133 m), it should give a good estimate of local basal topography.

The results show that one iteration of GIA has a significant effect on ice sheet configuration, and in this case increases the total volume by about 5.8% (Fig. 6). In addition, since the basal topography becomes more depressed towards the center of the ice sheet relative to the initial simulation, the modelled ice surface topography is lower and has a more gentle gradient. A second iteration of GIA had only a minor effect on the calculated ice sheet (0.4% increase in volume from the first iteration).

Additional tests by (Gowan, 2014) for the full deglacial Laurentide Ice Sheet showed that there is only a weak dependence on ice volume and Earth model. For three layer (lithosphere, upper mantle, lower mantle) Earth models, the ice volume varied most with changes in lower mantle viscosity at LGM extent, but the difference was less than 0.5% (though smaller ice sheets will have less dependence on the lower mantle). Towards the end of deglaciation, there was more dependence on upper mantle viscosity, but again, the volume difference was less than 0.5%. Though the volume was close to the same, there were slight differences in the distribution of ice, though not by more than 100 m in extreme cases. Therefore, the recommendation when creating an ice sheet model is to include at least one iteration of GIA, but the chosen Earth model is not as important.

### 5 Conclusions

ICESHEET 1.0 is a program that can quickly create models of paleo-ice sheets, with a given margin configuration and estimated basal shear stress. We have provided two proof of concept examples showing configurations of the modern Greenland Ice Sheet and the Barents Sea Ice Sheet at the LGM. It is recommended that at least one iteration of GIA is included to best characterize the thickness and ice surface topography. This software has been used to create a full late glacial GIA based ice sheet model of the western Laurentide ice sheet (Gowan et al., in review). It is ideal for producing ice sheet models that have minimal input assumptions, but are glaciologically plausible. A suite of





ice sheet models through a glacial cycle could be used as inputs for climate and ice sheet dynamics modelling that are independent of those parameters.

## 5.1 Code availability

The source code, licensed under GPL version 3, and Greenland Ice Sheet example are available
in the supplementary material. Software updates will be available on EJG's website (http://www. raisedbeaches.net).

*Acknowledgements.* The ICESHEET software was developed as part of a PHD project by EJG, and was funded by an ANU Postgraduate Research Scholarship. This study is funded as part of a Swedish Research Council FORMAS grant (grant 2013-1600) to Nina Kirchner. We thank Nina Kirchner for comments that improved this
manuscript. Computing resources used for the development of ICESHEET were provided by Terrawulf (Sambridge et al., 2009). We thank Anna Hughes for providing the DATED margin at 20 ka prior to its publication. We also thank Kurt Lambeck for allowing us to use the ANU model as a template for the Barents Sea ice sheet example. Figures were created using GMT (Wessel and Smith, 1991).



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





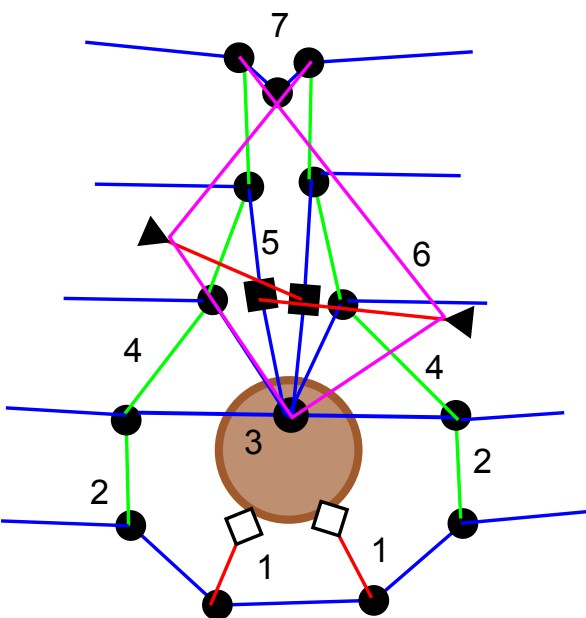

**Figure 1.** Schematic illustrating the steps in calculating the ice sheet. The blue lines represent a contour of equal ice surface elevation, with increasing elevation towards the top. 1) Calculated flowlines are flagged as they do not reach the next contour elevation due to a topographic barrier (represented by the brown circle). 2) Flowline calculation to the next contour is successful. 3) The points where flowline calculation is initiated along the contour polygon are resampled. Flowline calculation is not initiated for the point in the brown circle, as its elevation is too high. 4) Flowline calculation is successful for the points on either side of the topographic high. The point in the topographic high is not eliminated at this step. 5) After another flowline step, the distance along the flowline and the point within the topographic high is sufficient that the resampling puts points at this spot (black squares). 6) The flowline calculation causes the polygon to cross over itself. The polygon is isolated (magenta lines). Since the calculated direction of flow at these points is outside of this isolated polygon, it is eliminated. 7) Final contour polygon includes the crossover point.





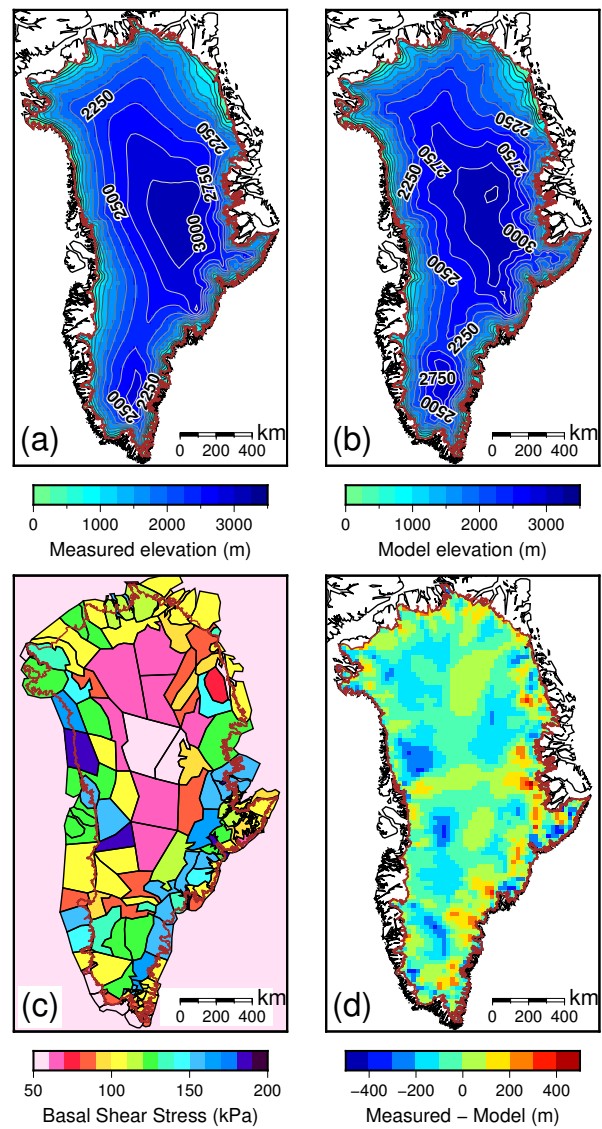

**Figure 2.** Sample model of the contemporary Greenland Ice Sheet. (a) Modern topography. The brown line is the current grounded ice margin, the black lines are the modern day coastlines. (b) Modelled topography. (c) Basal shear stress domains and values used to construct the model. (d) Difference between the observed topography and modelled topography.



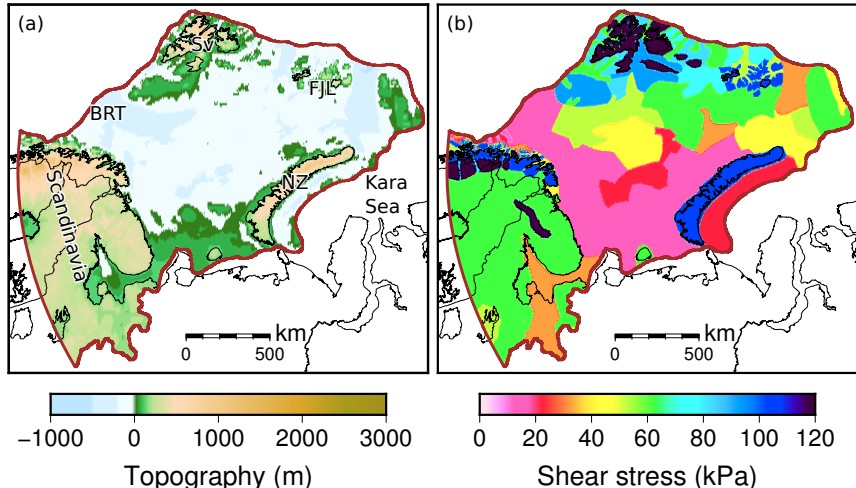

**Figure 3.** Basal topography used in the resolution test, which is modern topography minus the 133 m drop in global mean sea level at 20 ka. Also shown in brown is the 20 ka ice margin (Hughes et al., 2016) and the location of places described in the text. Sv - Svalbard. FJL - Franz Josef Land. NZ - Novaya Zemlya. BRT - Bear Island Trough. (b) Basal shear stress values used in the example in this paper.

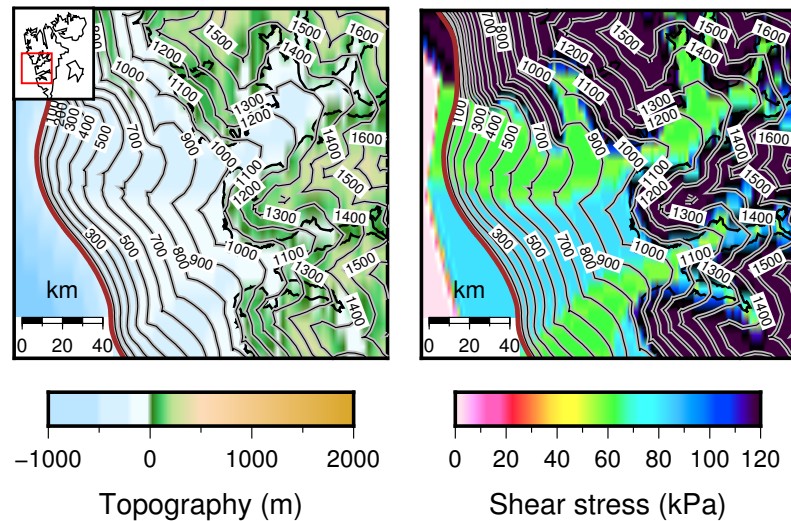

**Figure 4.** Example from central-western Svalbard of how changes in basal topography and basal shear stress affect the modelled ice surface topography of the ice sheet. Contour interval is 100 m in the figure, though this sample was calculated with a 5 km spacing and 20 m contour interval. The dark black lines are the modern day coastlines.



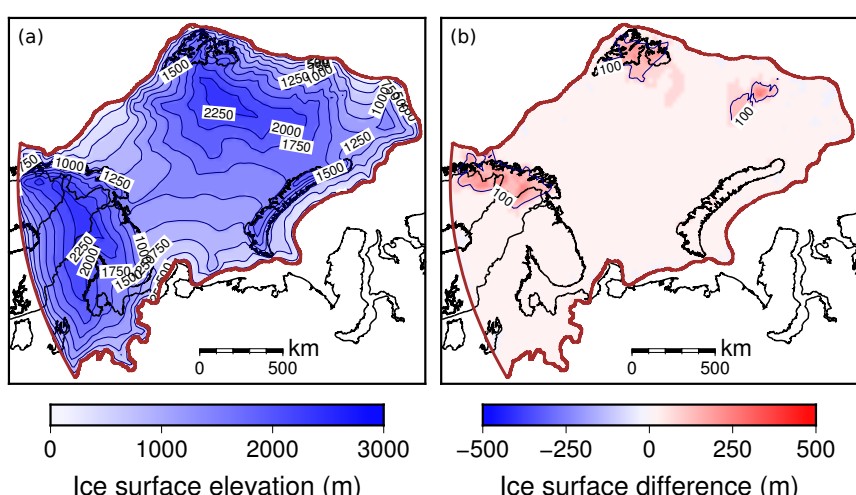

**Figure 5.** (a) Reference ice sheet model using the topography and basal shear stress in Fig. 3. (b) The difference between a model calculated with a 20 km spacing and 20 m contour interval and the reference model shown in (a). This demonstrates that the lower resolution tends to overestimate the ice surface elevation in mountainous regions.





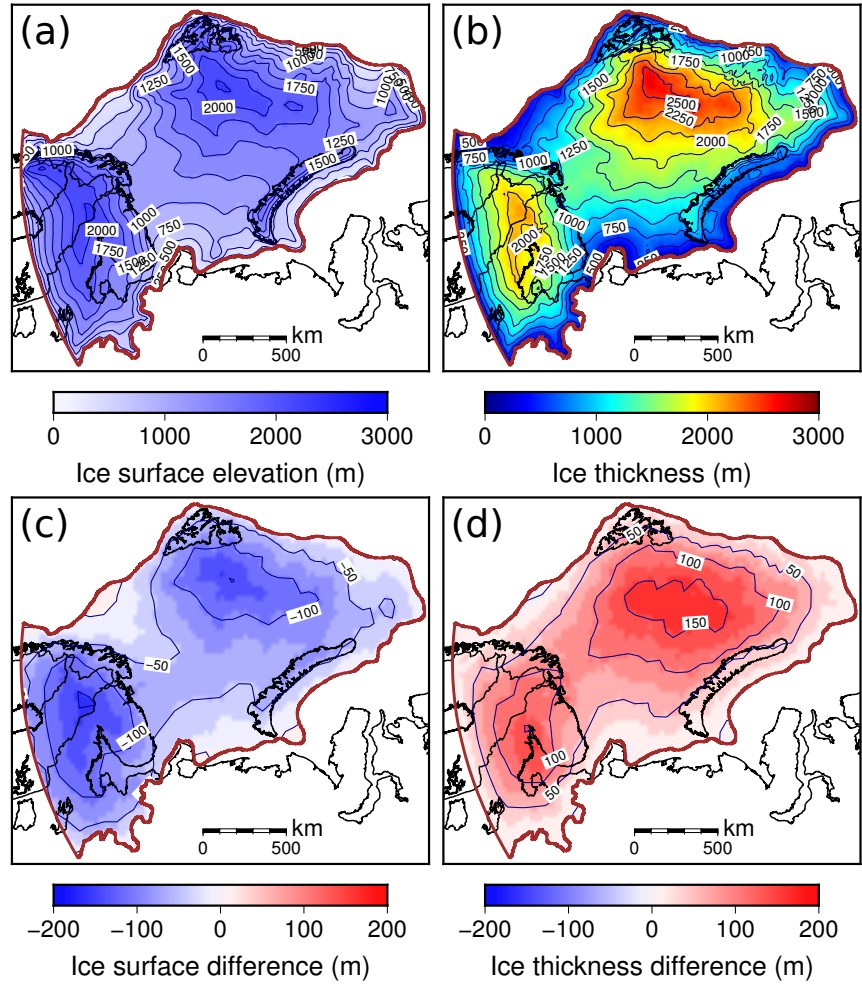

**Figure 6.** Ice sheet model after one iteration of GIA. (a) Ice surface elevation. (b) Ice thickness (c) Difference in elevation between (a) and the initial model without GIA deformed topography. (d) same as (c) but for ice thickness.



**Table 1.** Results of the resolution test

| Spacing (km) | Contour interval (m) | CPU Execution time (min)[1] | Ice Volume ($10^6$ km$^3$) | Element difference (%)[2] |
|---|---|---|---|---|
| 1 | 10 | 192.7 | 3.633 | 0.00 |
| 1 | 20 | 123.7 | 3.642 | 0.00 |
| 1 | 30 | 98.0 | 3.652 | 0.00 |
| 1 | 40 | 85.0 | 3.674 | 0.27 |
| 1 | 50 | 78.2 | 3.702 | 1.51 |
| 3 | 10 | 62.8 | 3.639 | 0.00 |
| 3 | 20 | 39.8 | 3.643 | 0.00 |
| 3 | 30 | 31.4 | 3.652 | 0.00 |
| 3 | 40 | 26.9 | 3.660 | 0.06 |
| 3 | 50 | 24.4 | 3.673 | 0.56 |
| 5 | 10 | 38.4 | 3.647 | 0.00 |
| 5 | 20 | 24.4 | 3.652 | 0.02 |
| 5 | 30 | 19.2 | 3.658 | 0.05 |
| 5 | 40 | 16.4 | 3.665 | 0.13 |
| 5 | 50 | 14.5 | 3.677 | 0.62 |
| 10 | 10 | 19.4 | 3.668 | 0.39 |
| 10 | 20 | 12.2 | 3.672 | 0.63 |
| 10 | 30 | 9.4 | 3.677 | 0.73 |
| 10 | 40 | 8.0 | 3.683 | 0.84 |
| 10 | 50 | 7.4 | 3.693 | 1.39 |
| 15 | 10 | 13.1 | 3.687 | 1.32 |
| 15 | 20 | 8.4 | 3.691 | 1.44 |
| 15 | 30 | 6.6 | 3.695 | 1.50 |
| 15 | 40 | 5.6 | 3.703 | 1.70 |
| 15 | 50 | 5.0 | 3.709 | 1.78 |
| 20 | 10 | 10.1 | 3.701 | 1.70 |
| 20 | 20 | 6.3 | 3.710 | 1.86 |
| 20 | 30 | 4.8 | 3.716 | 2.06 |
| 20 | 40 | 4.1 | 3.722 | 2.15 |
| 20 | 50 | 3.8 | 3.734 | 2.41 |
| 30 | 10 | 6.8 | 3.739 | 2.68 |
| 30 | 20 | 4.2 | 3.744 | 2.88 |
| 30 | 30 | 3.2 | 3.752 | 3.02 |
| 30 | 40 | 2.8 | 3.757 | 3.06 |
| 30 | 50 | 2.5 | 3.766 | 3.37 |

[1] Execution time on a Lenovo Thinkpad, Intel Core i7-4702MQ CPU at 2.20 GHz, on Linux Mint 17. Compiled with gfortran 4.8 with -O2 flag.

[2] Percent of 0.5° longitude by 0.25° elements that are > 100 m different from the reference model (out of 23205 total elements)