# Peer review of "ICESHEET 1.0: A program to produce paleo-ice sheet reconstructions with minimal assumptions"

_Geoscientific Model Development, 2016_

## Referee Comment (RC1) · Anonymous Referee #1 · 15 Feb 2016

The advance in science and computer power have led to an enormous increase in modelling techniques, number of models in computations, and complexity of models. Ice (sheet) models are input to several of modellings when it comes to e.g. climate change or sea-level rise. As the authors note, they also turn out to be more complex recently. I welcome this study as it (1) provides an easy tool for generating your own ice model and (2) shows that for many studies on glacial isostatic adjustment (GIA) the ice model does not have to be so complex as often indicated. The manuscript is well written and concise, figures and tables provide all information needed to understand the tool. While reading I made a few notes only where some additional information would help make points finally clear.

Minor suggestions: L109. Specify the "limited impact". L111. What is "a component of GIA"? L114. Add a few words on CALSEA such as underlying theory, resolution,

dimension, e.g. is it a spheric harmonic viscoelastic Maxwell body description up to degree 256 using PREM? Does it include rotational feedback and moving shorelines? 1-2 sentences here or in section 4.3. Sect. 3.2. Add maximum differences in the text. Aren't the narrow ice streams at the edge and in the fjord areas? L231/2. This sentences is out of place here. Suggest to remove and add to L226 "...were performed, whose results can be found in Table 1." Also add "(Table 1)" after polygons in L234. Figure 1. What are the red and green lines (name it in caption!)? Figure 4. Add ", see text" after ice sheet. Figure 5. Specify spacing and contour interval of the reference ice sheet model here as well. Table 1. Highlight/mark the reference model, the recommended result and the one used in Fig 5b.

---

## Referee Comment (RC2) · Anonymous Referee #2 · 19 Feb 2016

This paper describes an ice-sheet reconstruction technique, that estimates ice surface elevations given the 2-D margins of the ice sheet. It is based on two assumptions: steady state, and perfect plasticity, i.e., the basal shear stress is a given yield value. Both the bedrock topography and the basal yield stress (which can be uniform or spatially varying) need to be specified, but if they are not known, one or both can be adjusted iteratively to fit surface elevations (or ice sheet thicknesses) that may be known to some extent by independent means.

The reconstruction uses the method of characteristics, making direct use of two earlier studies: Reeh (1982) and Fisher et al. (1985). The authors have coded the method into a usable general-purpose program, which includes the capability to automatically correct for topographic barriers and contour crossovers. Although I have some reservations as described below, I think the study will be suitable for GMD, and the program

may be an interesting and worthwhile tool for some glaciological situations.

General comments:

Several significant aspects are not well explained, leaving some questions hanging, as described in points 1-3. Much of this can probably be corrected by clarification and additional text to provide more information and context.

1. The overall purpose and outcomes of the study and the program are not clearly defined. Given a map of ice-sheet margins, basically there are 3 unknowns: surface elevation, basal shear (yield) stress, and bedrock elevations (all 2-D maps). Eqs. 5 to 9 and the method of characteristics relate the latter two fields to the first. But if only one of the 3 is known or is available from some external means, the other 2 cannot both be determined uniquely. 2 of them have to be known to determine the 3rd. In the examples given, Greenland's bed topography and surface elevations are known from independent data, and the method determines (iteratively) the basal shear stress map. For the Barents, bed topography is given from a modern dataset (which neglects depression under paleo grounded ice), and basal stress is "adjusted" to yield GIA-estimated paleo ice thicknesses or equivalently surface elevations.

These applications confuse what many readers may assume up front is the main goal, which is to reconstruct paleo-ice sheet elevations (or thicknesses) given the margins. It would help to put all this in perspective, and to clarify the main outputs and purpose of the program.

The basic problem of constraining both basal topography and basal stresses crops up in many related papers, mostly using process-based physical ice sheet models (e.g., van Pelt et al., The Cryo., 2013 and references therein). Some perspective discussing the connection to these types of studies could be given.

2. Perhaps related to point # 1: if basal topography and/or basal stresses are unknown, and have to be iteratively adjusted so that the simulated surface elevations match those

Interactive
comment

from another data source, then what is learned from the exercise? In other words, if the surface elevations (or thicknesses) need to be known a priori, then why re-simulate them? There may be good reasons: perhaps to produce a higher-resolution surface elevation map, or to produce an iterated map of basal stresses which provides insights into bedrock geology or basal liquid/thermal regimes. But these issues are not clearly addressed and should be discussed clearly.

3. The basic sequence of using the method of characteristics to step elevation contours incrementally upwards from one contour to the next is clearly explained here, as it is in the previous papers. However, the procedures used to handle topographic barriers (nunataks), and crossovers in the contours, are very opaque. Fig. 1 does not explain them well. For instance, in Fig. 1 and its caption: - it would help to say that a topographic barrier is the result of E becoming < B (where E is ice surface elevation along a flowline and B is bedrock elevation). - What are the triangles and the orange lines in Fig. 1? - What is "resampling"? This should be described in more detail.

I suggest splitting Fig. 1 into two or even three separate figures: one for the basic stepping algorithm with no complications, one for a nunatak, and one for a crossover. This could greatly improve its potential to be understood.

Along with Fig. 1, the text describing some of the procedural steps in section 2.2 could be expanded and clarified, especially steps 6 and 7. Also, the connections between steps 1 to 14 in the text and the parts of the figure(s) could be specified more thoroughly.

4. The authors are probably well aware of the limitations of the basic assumptions of steady state and perfect plasticity, and that there are some ice sheet regions and intervals where they may be seriously in error (e.g., Laurentide during deglaciation, West Antarctic ice streams). These should be mentioned as caveats.

5. I think the term "ice-sheet models" in the title and text is misleading. To many readers, this implies process-based time-stepping ice sheet models based on conservation

equations of mass, momentum and heat. Instead I would suggest "reconstructions", as used for instance in Fisher et al.'s title ("...objective reconstructions").

Specific comments:

a. Text could be added to note how this study goes beyond Reeh (1982) and Fisher et al (1985), besides providing a program. For instance, bed topography can be iteratively adjusted here (pg. 4, line 110), rather than modifying the equations to represent isostatic equilibrium with the ice load.

b. It might be helpful to some readers to add hints for the derivation of Eq. 8 and 9, to give them a better chance of deriving them themselves if they are so inclined. The derivations of Eqs. 6 and 7 are straightforward, but Eq. 8 is more challenging. Just saying "As in the development of A6, one starts $dq/dx = ...$" in Fisher's Appendix would help a lot.

Also, a couple of features of the equations could be stated which, although fairly obvious, might be helpful to readers on first perusal: (1) The flowline direction determined by Eq. 6 is the direction of local steepest ascent of the ice surface. (2) If x, y, E, p, q are known at a point on a flowline (and B and Hf are known everywhere), then Eqs. 6 to 8 (or 9) yield the next y, E, q along the flowline for a given increment in x. And then the next p is known from Eq. 5.

c. In the Greenland example, it would be interesting to iterate on Greenland bed topography starting from a flat surface, instead of using modern bedrock data, and not iterate on basal stresses. That would be more analogous to a Laurentide application.

d. Does Fig. 2c show the initial basal stresses, or the final iterated values? (see pg. 7, line 183).

e. In Fig. 4, it is unclear what is plotted. The caption says "changes" in basal topography and shear stress, but the plots are absolute fields.

Technical corrections:

pg. 2, line 26: "each flowline ray *is* allowed..."

pg. 2, Eq. 2: Note that this is only true for flat bedrock, B=0.

pg. 4, line 105: What does "time interval" mean, given that everything is in steady state?

pg. 5, line 136: Instead of "too steep", "too high" would be more precise. And perhaps add "where E < B".

pg. 9, line 265: "dependence of ice volume on the Earth model..." (?)

―――――――――――――――

---

## Referee Comment (RC3) · Anonymous Referee #3 · 19 Feb 2016

Review of ICESHEET 1.0 Gowan et al. manuscript

This manuscript describes a simple ice sheet model which can be used to simulate the first order surface elevation of an ice sheet given its extent. The model would be useful to many studies reconstructing past ice sheets from dating the chronology or retreat or using Glacial Isostatic Adjustment modelling. The manuscript is well, written, and concise and is well suited to this journal. I think the manuscript would need a moderate amount of corrections before publication. In particular, the terminology used in the manuscript needs adjusting and the applications to the Greenland and Eurasian ice sheets need some more detail. One important test is the sensitivity to the model resolution. This is done on the Eurasian ice sheet where there is no observational data on ice thickness. The resolution tests should be instead done for the Greenland ice sheet.

Details comments:

- In the manuscript, the model is referred to as "a program" (title, abstract) "a numerical program" (l29) "modelling software" (l41) and "program" and "software" in the conclusion. "model" is used here to describe an ice sheet "simulation" or "reconstruction" I think that this terminology is confusing. It should be described as a "numerical model". You could also use the word "simulator" which some statisticians use to differentiate physical models from statistical models. If you have good reasons to stick to the terminology, please clarify the definitions you use.

- Similarly I would replace "modelling procedure" with "algorithm" or equivalent terminology

- Please indicate how this model compares with other similar models, not only in terms of the equations, but also in the solving procedure.

- Replace "sample model" in titles 3 and 4 with "Example" or "application"

- Section 3.1 how does the basal shear stress compare with other modelling studies of the Greenland ice sheet ? How much does it affect the results? This is important since the goal here is to comare the model results to observations.

- Section 3.2: Please compare the difference in ice sheet volume modelled vs estimates from observations.

- Please include resolution tests for the Greenland ice sheet.

- line 274 add "3D topographical" before "models of palaeo-ice sheets"

- line 282: "those parameters" : rephrase to make it clear what parameters you are talking about.

- In your conclusion, please mention the main results from the sensitivity tests presented here.

---

## Short Comment (SC1) · 16 Mar 2016

Most recent scientific work in modeling past ice sheets has been aimed at a simple or a complex endmember. The former includes whole-ice-sheet simple flowline modeling and the "ice-cream scoop" approach of the ICE-nG models, in which ice volume is "scooped" from the ocean and placed on the map in a way that is semi-arbitrary but fits the GIA constraints. The latter includes all attempts to use time-evolving ice-dynamics models.

Gowan et al. present work that is sorely needed, that obeys the physics without over-fitting the geological constraints. I am very enthusiastic about this work, and see this as a necessary way forward. More specifically, I think this work embodies the null hypothesis: ice sheets in the past behave as physics dictates. Modeling them in an

equilibrium state should be the zeroth- or first-order work that forms the basis for any more complex investigation, and has significant scientific value in and of itself.

A few minor comments follow.

- Elevation: $E$ should become $z_s$ (z-surface) or something with $z$ in it – $E$ to me is Young's modulus, erosion, ... while $z$ is a field vertical positions. Same goes with $B$, would be more intuitive to have $z_b$. Thus the equations with $E - B$ would become $z_s - z_b$, and this meaning would become immediately apparent to me.

- Line 66 – no comma needed

- $p$ and $q$: once again, for readability, I would suggest avoiding variables like $q$ that already mean something to glaciologists. Maybe some consecutive Greek letters or other ones from our standard alphabet would work. I don't mean to be a stickler about this – it's just that this makes the difference to me between being able to understand what you're doing after a skim, and after a close reading, and I think that anything that you do to increase the at-a-glance readability will increase the paper's impact.

- Your steps in working through the model are good. How about a flowchart to accompany this? I find these very useful, and use a program called yEd, which is pretty quick.

- Nice examples, especially illustrative of the importance of basal shear stress inputs.

- Software repository location: I would suggest that it could be useful to also provide the software on a non-personal website. This should increase its visibility and ensure its future availability. Some researchers like archives that have a doi, and GMD is in support of this. I personally use GitHub for everything, which
is nice because it allows others to follow changes to one's source code and/or check it out and modify it and suggest changes.

Overall, this work is elegant in its simplicity, and I look forward to seeing additional applications.

– Andy

---

## Author Comment (AC1) · 5 Apr 2016

[12pt]article

**1   Overview**

First, we would like to thank the reviewers who made comments. We have carefully considered all the points and made changes and additions that make the study more complete. One of the main changes we have made is to change the order of sections 3 and 4, which makes the paper flow better, with consideration of the resolution tests. Also, the terminology was made to be consistent throughout the paper, so that this is a

"program" that produces "ice sheet reconstructions". This has required that we revise the title of the paper to "ICESHEET 1.0: A program to produce paleo-ice sheet reconstructions with minimal assumptions" (previously titled "ICESHEET 1.0: A program to produce paleo-ice sheet models with minimal assumptions"). Figure 1 has been completely changed, and we hope it provides a more clear illustration of how the program works.

Below are the reviewer comments, followed by our response in italics. The revised manuscript with text changes is included as a separate document.

**2  Response to comments by Anonymous Referee #1**

The advance in science and computer power have led to an enormous increase in modelling techniques, number of models in computations, and complexity of models. Ice (sheet) models are input to several of modellings when it comes to e.g. climate change or sea-level rise. As the authors note, they also turn out to be more complex recently. I welcome this study as it (1) provides an easy tool for generating your own ice model and (2) shows that for many studies on glacial isostatic adjustment (GIA) the ice model does not have to be so complex as often indicated. The manuscript is well written and concise, figures and tables provide all information needed to understand the tool. While reading I made a few notes only where some additional information would help make points finally clear.

Minor suggestions:

L109. Specify the "limited impact".

**Response**: *The final results change by less than 100 m near the margins if the initial sea level is adjusted. This is noted in the text.*

L111. What is "a component of GIA"?

[Figure]

**Response**: *The wording there was awkward. We changed this to say "In subsequent iterations, the topography is adjusted for glacial-isostatic adjustment..."*

L114. Add a few words on CALSEA such as underlying theory, resolution, dimension, e.g. is it a spheric harmonic viscoelastic Maxwell body description up to degree 256 using PREM? Does it include rotational feedback and moving shorelines? 1-2 sentences here or in section 4.3.

**Response**: *We added the following sentences to explain how CALSEA works:*

*CALSEA computes glacial-isostatic adjustment using a spherically symmetric Earth, with a Maxwell rheology mantle and elastic lithosphere, using the PREM model (Dziewonski and Anderson, 1981) for other Earth model parameters. In includes time evolving shorelines and rotational feedback.*

Sect. 3.2. Add maximum differences in the text. Aren't the narrow ice streams at the edge and in the fjord areas?

**Response**: *We added a numerical value to the text (>400 m). Also added that indeed, the largest discrepancies are at the edge of the ice sheet sheet, where the ice stream locations are located.*

L231/2. This sentences is out of place here. Suggest to remove and add to L226 "...were performed, whose results can be found in Table 1." Also add "(Table 1)" after polygons in L234.

**Response**: *We rearranged the reference to Table 1 in the first paragraph of section 4.2 and added the reference to it in the first sentence of the second paragraph.*

Figure 1. What are the red and green lines (name it in caption!)?

**Response**: *Figure 1 has been completely revised. This is elaborated in response to Anonymous Referee #2.*

Figure 4. Add ", see text" after ice sheet.

**Response**: *This has been added to the caption.*

Figure 5. Specify spacing and contour interval of the reference ice sheet model here as well.

**Response**: *We added the information to the caption (1 km spacing and 10 m contour interval)*

Table 1. Highlight/mark the reference model, the recommended result and the one used in Fig 5b.

**Response**: *Notes have been added to not the reference and recommended reconstructions.*

**3   Response to comments by Anonymous Referee #2**

This paper describes an ice-sheet reconstruction technique, that estimates ice surface elevations given the 2-D margins of the ice sheet. It is based on two assumptions: steady state, and perfect plasticity, i.e., the basal shear stress is a given yield value. Both the bedrock topography and the basal yield stress (which can be uniform or spatially varying) need to be specified, but if they are not known, one or both can be adjusted iteratively to fit surface elevations (or ice sheet thicknesses) that may be known to some extent by independent means.

The reconstruction uses the method of characteristics, making direct use of two earlier studies: Reeh (1982) and Fisher et al. (1985). The authors have coded the method into a usable general-purpose program, which includes the capability to automatically correct for topographic barriers and contour crossovers. Although I have some reservations as described below, I think the study will be suitable for GMD, and the program may be an interesting and worthwhile tool for some glaciological situations.

General comments:

Several significant aspects are not well explained, leaving some questions hanging, as described in points 1-3. Much of this can probably be corrected by clarification and additional text to provide more information and context.

1. The overall purpose and outcomes of the study and the program are not clearly defined. Given a map of ice-sheet margins, basically there are 3 unknowns: surface elevation, basal shear (yield) stress, and bedrock elevations (all 2-D maps). Eqs. 5 to 9 and the method of characteristics relate the latter two fields to the first. But if only one of the 3 is known or is available from some external means, the other 2 cannot both be determined uniquely. 2 of them have to be known to determine the 3rd. In the examples given, Greenland's bed topography and surface elevations are known from independent data, and the method determines (iteratively) the basal shear stress map. For the Barents, bed topography is given from a modern dataset (which neglects depression under paleo grounded ice), and basal stress is "adjusted" to yield GIA-estimated paleo ice thicknesses or equivalently surface elevations.

These applications confuse what many readers may assume up front is the main goal, which is to reconstruct paleo-ice sheet elevations (or thicknesses) given the margins. It would help to put all this in perspective, and to clarify the main outputs and purpose of the program.

The basic problem of constraining both basal topography and basal stresses crops up in many related papers, mostly using process-based physical ice sheet models (e.g., van Pelt et al., The Cryo., 2013 and references therein). Some perspective discussing the connection to these types of studies could be given.

**Response**: *To address this, we have added a sentence to the third paragraph of the introduction:*

*"The goal of this software is to provide an compromise between the GIA-only ice sheet*

*reconstructions that have limited or no physics applied to their construction, and the full glacial systems models that demand considerable computational resources."*

*We also added at the end of that paragraph:*

*"Ultimately, the goal would be to reconstruct in a time-stepped fashion the entire history of an ice sheet complex. In this case, the basal topography is relatively well determined (since there is no existing ice), and the basal shear stress can be established to a certain extent by the surficial geology and geomorphology. The ice topography and basal shear stress are determined through time using external evidence, such as the nature of GIA. An example of this is presented for the western Laurentide Ice Sheet in Gowan et al (2016)."*

*As for the studies like the one by van Pelt et al (2013), they determine the basal parameters from direct observations of ice dynamics, something that is not really possible in a paleo-ice sheet (except perhaps from trying to match flowline patterns that happen to be preserved on the landscape from near the end of glaciation). The end goal of the Greenland example is basically to show that it is possible to reconstruct a modern ice sheet using our software, and applying GIA deformation is not necessary. That said, we now include reference and comparison to the results of these dynamic studies in section 3.*

2. Perhaps related to point # 1: if basal topography and/or basal stresses are unknown, and have to be iteratively adjusted so that the simulated surface elevations match those from another data source, then what is learned from the exercise? In other words, if the surface elevations (or thicknesses) need to be known a priori, then why re-simulate them? There may be good reasons: perhaps to produce a higher-resolution surface elevation map, or to produce an iterated map of basal stresses which provides insights into bedrock geology or basal liquid/thermal regimes. But these issues are not clearly addressed and should be discussed clearly.

**Response**: *The entire goal of this study is to present a way to produce basic paleo-ice*

[Figure]

*sheet models (for GIA modelling), and in most cases the ice sheet thickness is not known a priori, but basal topoography usually is. Although our software could be used to determine the basal characteristics of contemporary ice sheets, we have not and are not suggesting it be done (since higher order dynamic ice sheet modelling is possible in these cases). This was already stated explicitly in sections 1, 3 and 4, and are covered by the additional sentences added (as above).*

3. The basic sequence of using the method of characteristics to step elevation contours incrementally upwards from one contour to the next is clearly explained here, as it is in the previous papers. However, the procedures used to handle topographic barriers (nunataks), and crossovers in the contours, are very opaque. Fig. 1 does not explain them well. For instance, in Fig. 1 and its caption: - it would help to say that a topographic barrier is the result of E becoming < B (where E is ice surface elevation along a flowline and B is bedrock elevation). - What are the triangles and the orange lines in Fig. 1? - What is "resampling"? This should be described in more detail.

I suggest splitting Fig. 1 into two or even three separate figures: one for the basic stepping algorithm with no complications, one for a nunatak, and one for a crossover. This could greatly improve its potential to be understood.

Along with Fig. 1, the text describing some of the procedural steps in section 2.2 could be expanded and clarified, especially steps 6 and 7. Also, the connections between steps 1 to 14 in the text and the parts of the figure(s) could be specified more thoroughly.

**Response**: *We have redone figure 1 to specifically illustrate several of the steps explicitly, in particular steps 7 (stepping with no complications), 8 (hitting a nunatuk), 10, 11 (crossovers), 12 and 14 (resampling). The other steps basically relate to reading and writing files and the invocation of subroutines (i.e. steps 1-6).*

4. The authors are probably well aware of the limitations of the basic assumptions of steady state and perfect plasticity, and that there are some ice sheet regions and

intervals where they may be seriously in error (e.g., Laurentide during deglaciation, West Antarctic ice streams). These should be mentioned as caveats.

**Response**: *Indeed we are aware of this (and it was mentioned in the abstract), we have added the following to the end of section 2.1:*

*"It is important to note that assuming perfectly plastic, steady state conditions for the ice sheet is not accurate in areas where the ice sheet was highly dynamic. Due to this, the output basal shear stress is unlikely to reflect the true basal shear stress in those areas".*

5. I think the term "ice-sheet models" in the title and text is misleading. To many readers, this implies process-based time-stepping ice sheet models based on conservation equations of mass, momentum and heat. Instead I would suggest "reconstructions", as used for instance in Fisher et al.'s title ("...objective reconstructions").

**Response**: *We have changed the text to try and be consistent with the terminology throughout the paper in terms of calling these "reconstructions", including the title.*

Specific comments:

a. Text could be added to note how this study goes beyond Reeh (1982) and Fisher et al (1985), besides providing a program. For instance, bed topography can be iteratively adjusted here (pg. 4, line 110), rather than modifying the equations to represent isostatic equilibrium with the ice load.

**Response**: *We have added the following sentence to the end of section 2.1:*

*"In the next subsection, we note some of the improvements to the original methodology, including adjustments to the base topography with realistic GIA, dealing with margins that are in marine environments, automatic determination of ice sheet saddles, and adjusting for the presence of nunataks."*

b. It might be helpful to some readers to add hints for the derivation of Eq. 8 and 9,

to give them a better chance of deriving them themselves if they are so inclined. The derivations of Eqs. 6 and 7 are straightforward, but Eq. 8 is more challenging. Just saying "As in the development of A6, one starts dq/dx = ..." in Fisher's Appendix would help a lot.

**Response**: *These equations were derived explicitly (in a general sense) in Differential-gleichungen: Losungsmethoden und Losungen by Kamke (1965). A reference to that has been added.*

Also, a couple of features of the equations could be stated which, although fairly obvious, might be helpful to readers on first perusal: (1) The flowline direction determined by Eq. 6 is the direction of local steepest ascent of the ice surface. (2) If x, y, E, p, q are known at a point on a flowline (and B and Hf are known everywhere), then Eqs. 6 to 8 (or 9) yield the next y, E, q along the flowline for a given increment in x. And then the next p is known from Eq. 5.

**Response**: *On point (1), the following text has been added immediately after the set of equations:*

*"Equation 6 gives the direction of local maximum steepness."*

*As for point (2), although this is true, in general for a paleo-ice sheet we don't explicitly know what the values of E, p and q are beforehand, and if we did, there would be no point in going through this exercise.*

c. In the Greenland example, it would be interesting to iterate on Greenland bed topography starting from a flat surface, instead of using modern bedrock data, and not iterate on basal stresses. That would be more analogous to a Laurentide application.

**Response**: *In the Laurentide example (Gowan et al, 2016), the basal topography is the only piece of information that we can be certain of. We did not start with a flat topography in that study, nor did the study of Fisher et al (1985). If you started from a flat surface for Greenland, you would need additional information (i.e. known values of*

*basal shear stress) in order to uniquely determine the bedrock topography. We do not see the value of this kind of exercise since we are not aware of any studies where the basal shear stress has been determined for the entire ice sheet.*

d. Does Fig. 2c show the initial basal stresses, or the final iterated values? (see pg. 7, line 183).

**Response**: *This is the final iterated values. This is now indicated in the figure caption.*

e. In Fig. 4, it is unclear what is plotted. The caption says "changes" in basal topography and shear stress, but the plots are absolute fields.

**Response**: *This is refering to spatial changes in topography and basal shear stress. The word "spatial" has been added to the caption.*

Technical corrections:

pg. 2, line 26: "each flowline ray *is* allowed..."

**Response**: *The word "is" has been added to the text.*

pg. 2, Eq. 2: Note that this is only true for flat bedrock, B=0.

**Response**: *We included "neglecting basal topography" to the sentence after equation 2.*

pg. 4, line 105: What does "time interval" mean, given that everything is in steady state?

**Response**: *Yes, "interval" is an inaccurate term for this. We changed it to say "epoch".*

pg. 5, line 136: Instead of "too steep", "too high" would be more precise. And perhaps add "where E < B".

**Response**: *We changed that line to say "too high", and added E<B to the subsequent parenthesis.*

pg. 9, line 265: "dependence of ice volume on the Earth model..." (?)

**Response**: *We rephrased this sentence to say:*

*"...a weak dependence on reconstructed ice volume and Earth model used to compute GIA"*

**4   Response to comments by Anonymous Referee #3**

This manuscript describes a simple ice sheet model which can be used to simulate the first order surface elevation of an ice sheet given its extent. The model would be useful to many studies reconstructing past ice sheets from dating the chronology or retreat or using Glacial Isostatic Adjustment modelling. The manuscript is well, written, and concise and is well suited to this journal. I think the manuscript would need a moderate amount of corrections before publication. In particular, the terminology used in the manuscript needs adjusting and the applications to the Greenland and Eurasian ice sheets need some more detail. One important test is the sensitivity to the model resolution. This is done on the Eurasian ice sheet where there is no observational data on ice thickness. The resolution tests should be instead done for the Greenland ice sheet.

Details comments:

- In the manuscript, the model is referred to as "a program" (title, abstract) "a numerical program" (l29) "modelling software" (l41) and "program" and "software" in the conclusion. "model" is used here to describe an ice sheet "simulation" or "reconstruction" I think that this terminology is confusing. It should be described as a "numerical model". You could also use the word "simulator" which some statisticians use to differentiate physical models from statistical models. If you have good reasons to stick to the terminology, please clarify the definitions you use.
[Figure]

**Response**: *Throughout the text, we now refer to 'models' as "reconstructions. We also refer to it as a "program" throughout the entire text. The usage of the word "simulation" is no longer made when refering to our program.*

- Similarly I would replace "modelling procedure" with "algorithm" or equivalent terminology

**Response**: *We have replaced the section header for Section 2.2 (where this was used) to:*

*"Algorithm to reconstruct ice sheets"*

- Please indicate how this model compares with other similar models, not only in terms of the equations, but also in the solving procedure.

**Response**: *The Barents Sea reconstruction was based off the results of the ANU model, the methodology which was described in section 2.1.*

- Replace "sample model" in titles 3 and 4 with "Example" or "application"

**Response**: *We have changed this to say "Sample reconstruction"*

- Section 3.1 how does the basal shear stress compare with other modelling studies of the Greenland ice sheet ? How much does it affect the results? This is important since the goal here is to comare the model results to observations.

**Response**: *The goal here was not really to attempt to fully model the basal shear stress of the modern-day Greenland ice sheet, but rather to show that we can reconstruct a modern ice sheet reasonably well with the program using a similar process as with paleo-ice sheet reconstructions. As far as we are aware, there have only been a couple of studies that have attempted to determine the basal shear stress of the Greenland ice sheet, and those have been focused on small parts of the ice sheet where there is streaming ice (i.e. where the basal shear stress can be determined through the inversion of surface velocity values). Reference to these studies has been*

*added to the text:*

*Direct inversions for basal shear stress have only been performed for some of the ice streams in (e.g. Sergienko et al. 2014 and Shapero et al. 2016). In the study by Sergienko et al (2014), the basal shear stress exhibited a banded pattern, alternating between low (<50 kPa) to high (>150 kPa) values over spatial ranges of 5-20 km. Shapero et al (2016) found that the basal shear stress directly under fast flowing ice streams was almost negligible, but at the sides it could exceed 375 kPa. If averaged over a larger area, these values are consistent with the 100-200 kPa values in our reconstruction.*

- Section 3.2: Please compare the difference in ice sheet volume modelled vs estimates from observations.

**Response**: *We have added a couple of sentences on this (the volume values from the reconstructions are in Table 1)*

*The volume of the Greenland Ice Sheet, taken directly from the dataset by Morlighem et al (2014) is about $2.96 \times 10^6$ km$^3$. From Table 1, the reconstructed volume is within 5% of this value, except in the lowest resolution tests.*

- Please include resolution tests for the Greenland ice sheet.

**Response**: *Resolution tests have been added to table one for the Greenland ice sheet. A paragraph has been added to elaborate on the:*

*The resolution test was also performed with the Greenland simulation (Table 1). In this sample, the 5 km distance interval, 20 m contour interval does not perform quite as well as in the Barent Sea example. This is a result of having a larger area of mountainous terrain. Still, less than 3% of the elements are greater than 100 m different from the reference reconstruction, using a smaller spacing value may not be worth the extra computation time. If the area of focus is predominantly mountainous, it may be prudent to decrease the distance interval.*

- line 274 add "3D topographical" before "models of palaeo-ice sheets"

**Response**: *We have changed this sentence to say "reconstructions of paleo-ice sheets" to be consistent with previous terminology.*

- line 282: "those parameters" : rephrase to make it clear what parameters you are talking about.

**Response**: *We rephrased the final sentence to say:*

*"A suite of ice sheet reconstructions through a glacial cycle could be used as independent inputs for climate and ice sheet dynamics modelling."*

- In your conclusion, please mention the main results from the sensitivity tests presented here.

**Response**: *We added the sentence:*

*"It is also recommended (if a 5 km basal topography grid is used) to use a flowline spacing interval of 5 km and contour interval of 20 m for optimal calculation speed."*

**5  Response to comments by Andy Wickert**

Most recent scientific work in modeling past ice sheets has been aimed at a simple or a complex endmember. The former includes whole-ice-sheet simple flowline modeling and the "ice-cream scoop" approach of the ICE-nG models, in which ice volume is "scooped" from the ocean and placed on the map in a way that is semi-arbitrary but fits the GIA constraints. The latter includes all attempts to use time-evolving ice-dynamics models.

Gowan et al. present work that is sorely needed, that obeys the physics without overfitting the geological constraints. I am very enthusiastic about this work, and see this

as a necessary way forward. More specifically, I think this work embodies the null hypothesis: ice sheets in the past behave as physics dictates. Modeling them in an equilibrium state should be the zeroth- or first-order work that forms the basis for any more complex investigation, and has significant scientific value in and of itself.

A few minor comments follow.

Elevation: E should become z s (z-surface) or something with z in it – E to me is Young's modulus, erosion, ... while z is a field vertical positions. Same goes with B, would be more intuitive to have z b . Thus the equations with E – B would become z s – z b , and this meaning would become immediately apparent to me.

**Response**: *The notation used in this paper is identical to that of the original studies by Reeh and Fisher et al. (as noted in the text). We have chosen to keep that notation for consistency.*

Line 66 – no comma needed

**Response**: *The comma at line 66 has been removed.*

p and q: once again, for readability, I would suggest avoiding variables like q that already mean something to glaciologists. Maybe some consecutive Greek letters or other ones from our standard alphabet would work. I don't mean to be a stickler about this – it's just that this makes the difference to me between being able to understand what you're doing after a skim, and after a close reading, and I think that anything that you do to increase the at-a-glance readability will increase the paper's impact.

**Response**: *As above, the notation is the same as what was used in the derivative studies, and we have kept this for consistency. The p and q notation is actually from the PDE solutions from Kamke (1965).*

Your steps in working through the model are good. How about a flowchart to accompany this? I find these very useful, and use a program called yEd, which is pretty quick.

**Response**: *The program mostly works in a serial order as noted in section 2.2 (aside from the main contour loop), so we don't feel a flowchart would improve readabilty enough to justify inclusion. The revised figure 1 should give a better indication of what is involved in each step.*

Nice examples, especially illustrative of the importance of basal shear stress inputs.

Software repository location: I would suggest that it could be useful to also provide the software on a non-personal website. This should increase its visibility and ensure its future availability. Some researchers like archives that have a doi, and GMD is in support of this. I personally use GitHub for everything, which is nice because it allows others to follow changes to one's source code and/or check it out and modify it and suggest changes.

**Response**: *We are including the source code as a supplement to the paper. If future changes are required to the program, we will consider adding a Github or equivalent respoitory.*

Overall, this work is elegant in its simplicity, and I look forward to seeing additional applications.

**Supplement:**

[revised manuscript text omitted]

---

## Author Response (AR2)

**Topical Editor Decision: Publish subject to minor revisions (Editor review)** (20 Apr 2016) by Dan Goldberg
Comments to the Author:
I have read over the referee reports and the authors' response. all referees supported the work and had mainly technical comments and helpful suggestions. I am happy that the author has taken suggestions and comments on board.

I would like to make one more editorial suggestion before publication, regarding step 7 of section 2.2 and the corresponding subfigure in Fig 1. (I had hoped a referee would make mention, but either all referees were OK with this, or I am missing something, but it is a very minor change i suggest.)

Ostensibly a given elevation contour is meant to be just that, i.e. an isoline of elevation. if i understand the algorithm correctly, if you begin with an elevation contour and advance to the next, barring any "exceptions" (all other cases in Fig 1) the next contour would be an isoline of elevation as well. So what can lead to a contour having variable elevation, as implied by Fig 1(a) and step 7? Does this only occur when the starting contour is at the margin, and a constant thickness (1m) and variable elevation necessarily leads to this contour not being an isoline of elevation? I think just a sentence could be included for verification.

Response:

Thanks for for the suggestion. The original version of figure 1 described this, which is why it was not picked up by the referees. We forgot to include this description in the revised paper. We have added the following line to step 7:

[revised manuscript text omitted]